# A shared decision-making intervention between health care professionals and individuals undergoing Pulmonary Rehabilitation: An iterative development process with qualitative methods

**Amy C. Barradell**[1,2,3]*, **Hilary L. Bekker**[4,5], **Linzy Houchen-Wolloff**[1,2], **Kim Marshall-Nichols**[2], **Noelle Robertson**[6], **Sally J. Singh**[1]

1 Department of Respiratory Sciences, University of Leicester, Leicester, United Kingdom, 2 Centre for Exercise and Rehabilitation Science, Leicester Biomedical Research Centre-Respiratory, Glenfield Hospital, Leicester, United Kingdom, 3 National Institute for Health Research (NIHR) Applied Research Collaboration (East Midlands), College of Medicine, Biological Sciences & Psychology, Leicester General Hospital, Leicester, United Kingdom, 4 Leeds Unit of Complex Intervention Development (LUICD), Leeds Institute of Health Sciences—School of Medicine, University of Leeds, Leeds, United Kingdom, 5 Research Centre for Patient Involvement, Department of Public Health, Aarhus University, Aarhus, Denmark, 6 School of Psychology and Vision Sciences, University of Leicester, Leicester, United Kingdom

* Amy.Barradell@uhl-tr.nhs.uk

## Abstract

### Background

Pulmonary Rehabilitation (PR) services typically offer programmes to support individuals living with COPD make rehabilitation choices that best meet their needs, however, uptake remains low. Shared Decision-Making (SDM; e.g., Patient Decision Aids (PtDA)) interventions increase informed and values-based decision-making between individuals and healthcare professionals (HCPs). We aimed to develop an intervention to facilitate PR SDM which was acceptable to individuals living with COPD and PR HCPs.

### Methods

An iterative development process involving qualitative methods was adopted. Broad overarching frameworks included: complex intervention development framework, the multiple stakeholder decision making support model, and the Ottawa Decision Support Framework. Development included: assembling a steering group, outlining the scope for the PtDA, collating data to inform the PtDA design, prototype development, alpha testing with individuals with COPD (n = 4) and PR HCPs (n = 8), PtDA finalisation, and design and development of supporting components. This took nine months.

### Results

The PtDA was revised six times before providing an acceptable, comprehensible, and usable format for all stakeholders. Supporting components (decision coaching training and

**Data Availability Statement:** All relevant data are within the manuscript and its Supporting Information files.

**Funding:** This study contributes to ACB's PhD. It is funded by the National Institute for Health Research (NIHR) Applied Research Collaboration (ARC: East Midlands) and the Centre for Exercise and Rehabilitation Science (CERS) at the University Hospitals of Leicester NHS Trust. SJS is ACB's primary PhD supervisor and a Senior Investigator for the NIHR. The study was supported by the Leicester NIHR Biomedical Research Centre. The views expressed are those of the author(s) and not necessarily those of the NIHR or the Department of Health and Social Care. The funders had no role in study design, data collection and analysis, decision to publish, or preparation of the manuscript.

**Competing interests:** The authors have declared that no competing interests exist.

a consultation prompt) were necessary to upskill PR HCPs in SDM and implement the intervention into the PR pathway.

## Conclusions

We have developed a three-component SDM intervention (a PtDA, decision coaching training for PR healthcare professionals, and a consultation prompt) to support individuals living with COPD make informed and values-based decision about PR together with their PR healthcare professional. Clear implementation strategies are outlined which should support its integration into the PR pathway.

## Introduction

Living with Chronic Obstructive Pulmonary Disease (COPD) is known to impact an individuals' functional, psychological and social wellbeing [1–3]. One of the key treatment priorities for COPD is Pulmonary Rehabilitation (PR) [4], a programme which seeks to increase an individual's functional capacity and optimise their self-management behaviours. Traditionally PR is delivered face to face in a hospital or community setting over 6–8 weeks, with at least 2 supervised sessions per week. The programme consists of both resistance and endurance exercise, in addition to self-management education. There is strong evidence to show it improves individuals' breathlessness, emotional functioning, exercise capacity and perceived sense of control over their health [5].

Recently, alternative low cost home-based, web-enabled, and telerehabilitation models have been developed to support wider access of PR [6, 7]. In the COPD population there is growing evidence that home-based [8, 9], web-enabled [10], and telerehabilitation models [11, 12], have the potential to produce similar outcomes in exercise capacity, quality of life and perceived breathlessness when compared to a traditional programme [8, 10, 13].

Our PR service (Leicester Hospitals, UK) offers two home-based programmes for individuals living with COPD: a standardised COPD self-management manual (SPACE for COPD) which includes telephone support from healthcare professionals [14], and a comparable programme delivered online [12]. SPACE for COPD is evidenced in improving symptoms and exercise tolerance when compared to usual care [15], and is non-inferior to traditional PR for improvements in quality of life [9]. The online programme demonstrates potential for increasing disease knowledge and PR completion in a digitally literate subset of individuals post hospitalisation [16]. Both programmes are endorsed by individuals with COPD and PR healthcare professionals (HCPs) [17, 18].

During the Coronavirus Disease 2019 pandemic (Covid-19) there was an increased need for alternatives to traditional PR to support individuals living with COPD access a programme. However, the national audit continued to report disproportionate attendance to traditional PR and overall uptake as below target [19]. Commonly cited barriers included organisational constraints [e.g. limited timings/ locations for PR [20]], individuals' beliefs [e.g. reduced self-worth [21]], and socioeconomic status [e.g. cost implications for attending PR [22]]. To combat such barriers, guidance suggests that HCPs should support individuals living with COPD to make informed decisions between PR programme options [20, 23].

Shared Decision Making (SDM) interventions enable HCPs to support individuals to make informed decisions between healthcare options. The SDM process seeks to ensure the evidence-based data about healthcare options, their consequences for the health problem and

their impact on an individuals' life are discussed when individuals and HCPs share their perspectives of the health problem and the menu of options available. This enables individuals to consider all available options in turn and navigate to the option which is most appropriate and acceptable to their needs [24]. To facilitate SDM, Patient Decision Aids (PtDA) are frequently used [25, 26]. One way to integrate PtDAs within healthcare is by providing them prior to or during a healthcare consultation [26–30]. The consultation provides an established interaction point for individuals and HCPs to exchange their knowledge about the options, talk about their preferences, reason between the options in relation to their scientific and attributional data, and negotiate, tailor and plan care accordingly [31]. Integrating PtDAs within a service pathway has led to improvements in the frequency of SDM between individuals and HCPs and thereby the frequency of shared treatment decisions, particularly by increasing individuals' knowledge, increasing perceptions of risk, reducing any feelings of internal conflict, and reducing feelings of passivity [32–36]. Therefore, we might expect a SDM intervention to support individuals living with COPD and PR HCPs to make individualised treatment decisions about PR.

Currently there are no interventions to support individuals with COPD make decisions about PR [37] and so this intervention was developed to address this gap in research and service provision. The science of developing complex healthcare interventions is advancing as demonstrated by the presence of robust frameworks and guidelines [38–41]. Recently, the focus has shifted towards intervention implementation, namely the transparent account of intervention components and the inclusion of relevant stakeholders to support the smooth integration of interventions into healthcare contexts should they prove acceptable and effective.

This article describes the development of a SDM intervention for individuals living with COPD who are making PR treatment decisions with a PR HCP and illustrates how we involved expert stakeholders to consider the implementation of the intervention throughout its development. Our aim was to identify evidence to support the development of components within the intervention and its integration within the PR care pathway.

This manuscript has been written in accordance with the Guideline for reporting for intervention development studies (GUIDED) checklist) [42] to ensure a full and transparent account is provided.

## Methods

### Research ethics approval

This research was given ethical approval by South Leicester–Research Ethics Committee, reference 21/EM/0084.

The ethics committee waived the requirement for informed consent for participation in the intervention development work.

### Intervention development frameworks

The template for the reporting of PtDA development [41, 43] structures this section.

An iterative intervention development process involving qualitative methods was adopted to inform the components, content, structure, and integration of the SDM intervention into the PR care pathway. This process is reported in full in the protocol [44]. Briefly, these broad frameworks guided its development:

- The Medical Research Council (MRC) complex intervention development and evaluation framework provided overall methodological considerations for the different stages of intervention development, implementation, and testing [40].

- The Making Informed Decisions Individually and Together model (MIND-IT) [31] provided guidance for the development of intervention components at three levels: the individual, the HCP, and the consultation.

- The Ottawa Decision Support Framework (ODSF) [45] provided guidance for the identification of stakeholders needs and informed the content and implementation of the intervention.

Additional tools and frameworks were drawn upon to provide specific methodological guidance for developing PtDAs and to inform the development of intervention components. These are described alongside the intervention development steps below. Development began in March 2021 and was completed in December 2021.

## Assembling a steering group

To capture the voices of key stakeholders, individuals with COPD, PR specialists, PR service managers, and experts in SDM were invited to co-design the SDM intervention and implementation strategies.

## Outlining the scope of the SDM intervention

The steering group began by discussing the overall aim of the SDM intervention and preliminary specifications and content for the PtDA to inform its design. These discussions were informed by the user-centred design checklist [41] which provided robust guidance on the specific roles of all stakeholders in the development process.

## Collating data to inform the design of the PtDA

Fig 1 shows the use of the broad developmental models and frameworks which informed the PtDA prototype design. Design methods are presented in the Supplementary Materials (S1 in S1 File). These are referenced in Fig 2 as 'Prior to step 1.' Outputs from the design steps (Supplementary Materials, S2 in S1 File) informed the production of the PtDA prototype.

## PtDA prototype development

The Ottawa PtDA Development e-training (Available from: https://decisionaid.ohri.ca/eTraining/) was utilised to structure the development process. Fig 2 illustrates this process. The content of each step is detailed below.

**Step 1: Drafting the PtDA.** The PtDA was drafted using the Ottawa PtDA template to structure the decision-making process and guide the individual to communicate their informed preferences with others (e.g., HCPs, family, friends). Contextual information was added to tailor the PtDA to making decisions about PR (i.e., stating who the PtDA was for, what the decision to be made was, what the available options were and what factors might influence an individuals' decision-making). To structure the contextual data to describe COPD, the Common Sense Model of Self-Regulation [46] was used to ensure all aspects of individuals' illness representations (i.e. beliefs and expectations about an illness and its symptoms) were included.

**Step 2: Presenting information and re-drafting.** The PtDA was redrafted to ensure the information presented was comprehensive [47], balanced (i.e. presented without bias towards an option [48], and appropriate for the health literacy of the target population (i.e. <14 years old) [49–51]. Authors' conflicts of interests were added [52].

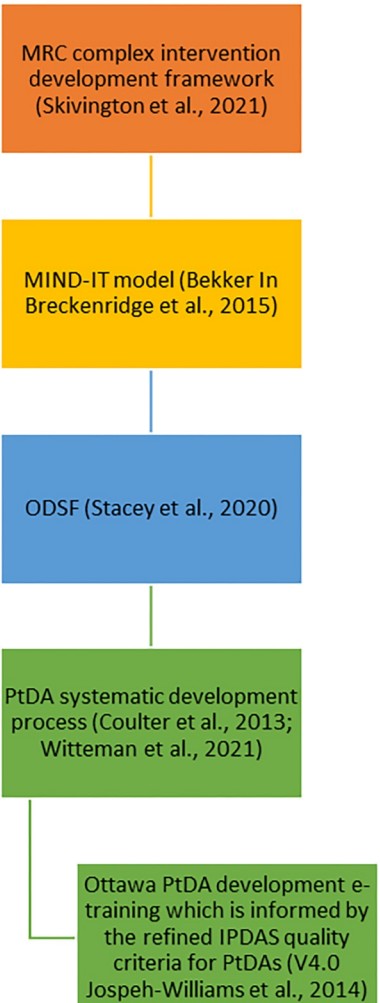

**Fig 1. Broad overarching models and frameworks used to create the PtDA prototype.** MRC = Medical Research Council, MIND-IT model = Making Informed Decisions Individually and Together model, ODSF = Ottawa Decision Support Framework, PtDA(s) = patient decision aid(s), IPDAS = International Patient Decision Aid Standards.

**Step 3: Use evidence-informed resources to present information about the options.** Comprehensive, critically appraised, and where available, characteristics and attributes based on research evidence for each of the healthcare options were added to the PtDA [47]. Data presentation was informed by Bonner and colleagues proposals to reduce the reader's cognitive load [53].

**Step 4: Clarifying values.** An exercise to support individuals in identifying their core values and beliefs was added to enable them to engage in values-based deliberation and decision-making [54]. This involved a process of identifying what matters to an individual relevant to their given health decision (i.e. PR decision). Patients used a section of the PtDA to complete 'What is important to me in my daily life?' and were asked to rate on a scale of 1–5 from 'not important' to 'extremely important' items such as 'daily activities in the home', 'work', spending time with family', 'understanding my COPD' etc.

**Step 5: Evaluating decision-making progress.** Components to measure the impact of the PtDA in supporting an individuals' decision-making were added. Whilst there is no consensus

**Prior to step 1**

- Synthesis of data from Design steps 1, 2a, 2b, 3, and 4 to inform the need for a SDM intervention involving a PtDA

**Step 1**

- Drafting the PtDA (version 1)
- Addition of contextual information to tailor the PtDA to making decisions about PR

**Step 3**

- Addition of comprehensive and critically appraised data for each treatment option
- Addition of a statement on the quality of data presented
- Versions 2-5

**Step 2**

- Redrafting the PtDA to ensure its ease of comprehension, balance and appropriateness for the health literacy of the COPD population
- Versions 2-5

**Step 4**

- Addition of a values-clarifcation exercise
- Versions 2-5

**Step 5**

- Addition of measures to evaluate the decision-making process and quality of decision made (SURE test, Measure of health literacy)
- Versions 2-5

**Step 6**

- Development of supporting intervention components (decision coaching training for PR HCPs, consultation prompt).

**Alpha testing**

- PtDA (version 5) testing with HCPs and individuals living with COPD
- Revisions and finalisation of PtDA (version 6)

**Fig 2. PtDA prototype development process.** SDM = shared decision making, PtDA = patient decision aid, PR = pulmonary rehabilitation, COPD = chronic obstructive pulmonary disease, HCPs = healthcare professionals.

or standardisation on the measures for evaluating this, reviewing the decision-making process and the quality of the decision that is made is recommended [35].

## Alpha testing

Alpha testing with individuals with COPD and PR HCPs was conducted to evaluate the comprehensibility, usability, and acceptability of implementation of the PtDA prototype (version 5) and produce the finalised PtDA (version 6). The Theoretical Framework of Acceptability was used to guide interpretations of the alpha testing [55].

Face to face testing was conducted with PR HCPs and then virtual testing was conducted with individuals from a local Patient and Public Involvement (PPI) group. One week prior to the meetings all were sent the PtDA prototype and asked to spend time reviewing it. During the meetings, we used a 'think-aloud approach' to encourage the PR HCPs and PPI group to share their thoughts of the PtDA whilst using it [56]. E.g. on the topic of 'Deciding between COPD management options', the patient group recommended changing the question to "Are there any questions you want to ask a PR HCP at your appointment?"

**Analysis and revisions.**   All observations from the meetings were recorded, collated and shared with the steering group. Each item was discussed in turn to make decisions about the actions required to address the proposed observations and amends. Agreed amends were made to finalise the PtDA. An iterative approach was adopted to re-draft and review the PtDA with the PPI and steering groups.

**Step 6: Supporting intervention components.**   Components to further support the implementation of the PtDA were added and their scope outlined. The steering group collectively reviewed information obtained during design steps (2b and 3, see Supplementary materials, S2 in S1 File) and alpha testing to inform this. Components included a consultation prompt and decision coaching training for PR HCPs.

In addition to the overarching intervention development frameworks [31, 40, 45], we drew upon guidance from the decision coaching framework [57]. The contribution of these models and frameworks are visualised in Fig 3. A member of the PR team was invited to contribute to the development of these components.

The decision coaching training content and activities (including the consultation prompt) were informed by proposed methods for teaching SDM [58] and the National Institute for Health and Care Excellence SDM learning package [59]. Guidance was sought on the best techniques to teach communication skills to HCPs [60–62]. To demonstrate SDM in practice and provide a robust step by step process for HCPs to adhere to, the three-talk model of SDM was selected [29].

A first draft of the decision coaching training plan and teaching materials (i.e., an example consultation prompt), were created and shared with the steering group for feedback. Their amends informed the finalised supporting intervention components.

## Results

### The steering group

All those approached agreed to join the steering group; individual living with COPD (n = 1), clinical academic specialist in pulmonary and cardiac rehabilitation (n = 2), specialist in clinical and health psychology (n = 1), trainee health psychologist (n = 1), and academic specialist in medical and SDM (n = 1). Formal online group meetings were held every 6–8 weeks to review and inform the development process.

For the development of supporting intervention components, a PR HCP (n = 1) agreed to join the steering group.

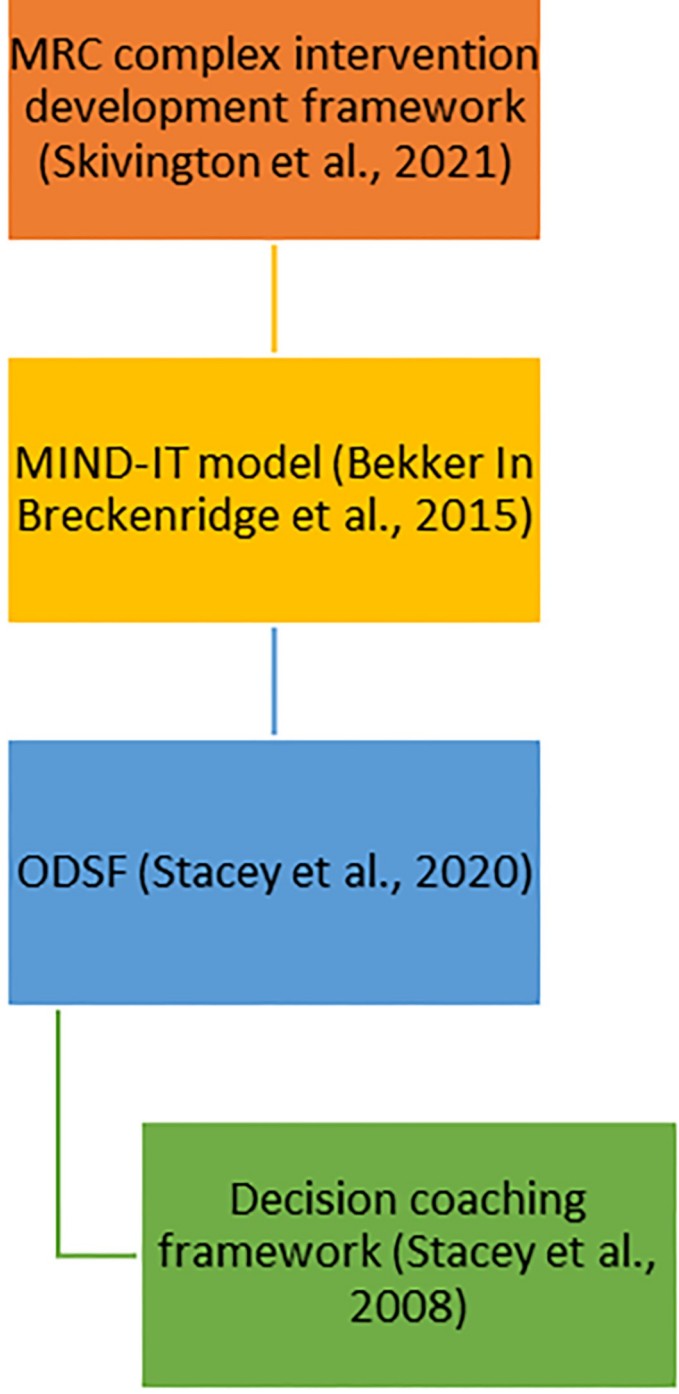

**Fig 3. Models and frameworks involved in the development of the decision coaching training and consultation prompt.** MRC = Medical Research Council, MIND-IT model = Making Informed Decisions Individually and Together model, ODSF = Ottawa Decision Support Framework.

No members of the steering group had a conflict of interest throughout the development, delivery, or evaluation of this intervention.

## The scope of the SDM intervention

The steering group agreed the scope of the SDM intervention was to standardise discussions of the menu of PR options between individuals and PR HCPs and support information exchange, deliberation, and decision-making.

The scope of the PtDA was to facilitate individuals' engagement with this SDM process. The intention was for it to be given to individuals upon their referral to PR to enable the opportunity for deliberation between the available options prior to and during their appointment with a PR HCP.

The scope of the decision coaching training was to facilitate HCPs' skills in guiding individuals' decision-making and implementing the chosen option. It aimed to develop PR HCPs' understanding, appetite, and acceptance for SDM and provide the necessary skills for using the PtDA in a PR consultation.

## Data to inform the design of the PtDA

These results of the design steps are presented in the Supplementary Materials (S2 in S1 File).

## The PtDA prototype

At the end of step five the prototype PtDA (version 5) had been produced. This was a 13-page interactive paper-based booklet informed by the results from the design steps and thereby personalised to individuals living with COPD who were making a decision about the menu of PR options. It had a readability score of 6.4 (appropriate for 11–12 years).

## Alpha testing

The meeting with PR HCPs was held on 24th August 2021. 7 individuals attended (2 males, 5 females) and one further individual provided feedback via email (1 female). The meeting with individuals living with COPD was held on 24th September 2021. 3 individuals attended (0 males, 3 females). 1 individual was unable to attend the meeting but provided feedback via email instead (1 male).

PR HCPs identified five areas of the Theoretical Framework of Acceptability that required attention: intervention coherence, affective attitude, burden, perceived effectiveness, ethicality, and self-efficacy (see Supplementary Materials; S3 in S1 File). The PPI group identified five branches of the Theoretical Framework of Acceptability that required attention: intervention coherence, affective attitude, perceived effectiveness, self-efficacy, and ethicality (see Supplementary Materials; S3 in S1 File).

There were high levels of agreement between the PPI group and PR HCPs regarding the value of the PtDA and the need for complimenting intervention components (i.e., the decision coaching training and the consultation prompt). Consequently, these components were developed (see Step 6: Supporting intervention components).

The PPI group and HCPs largely agreed that more information was required to illustrate the attributes of each option in the PtDA. We made the following modifications; PR HCPs wanted it to be clearer what the expectations were for individuals exercising at home, who the cohort of individuals within the centre-based option would be, and what the disadvantages of exercise testing include, and the PPI group wanted it to be clearer exactly when they would be likely to start each option, where each option would take place and with whom (i.e. with those

local to them), what the "complications" of exercise testing included, and what resources were needed to access each option.

PR HCPs and the PPI group expressed varying viewpoints on the use of certain terminology within the PtDA and the use of scientific data and references. Since we met with PR HCPs first, we were able to take their concerns to the PPI group so these could be addressed. This led us to make informed decisions on the retention or modification of these items within the PtDA.

Independent viewpoints included HCPs concerns about the time it would take to conduct SDM using the PtDA. We decided to address this concern in the decision coaching training for PR HCPs (see Step 6: Supporting intervention components).

For the usability of the PtDA, all of the PPI group's and PR HCPs' proposals were actioned. This led the PtDA to have clearer instructions (e.g., by changing "check" to "tick" throughout), additional prompts to encourage support from others in individuals' decision-making, activities that were easier to engage with (e.g., the addition of rating scales throughout), and a more digestible format (e.g., the orientation of the tables).

Proposals that were not implemented into the finalised PtDA, following discussion with the steering group, included the suggestion to add illustrative pictures to each option, the addition of the Covid-19 rehabilitation programme as a viable option, the addition of a rating scale for the 'Everyone's experience of COPD is different' section, addition of support from external groups (e.g. social/sport groups. . .) as an answer to the 'What is important to me in my daily life?' section, and the addition of a question about social or fitness groups to the 'Step 4: What are the next steps?' section. More detail of our reasoning for this is found in Supplementary Materials (S3 in S1 File).

## Finalised PtDA

The finalised PtDA [Version 6; a small excerpt (is provided in Supplementary Materials, S4 in S1 File] had a readability score of 7.6 (suitable for 12–13 years and thereby suitable for individuals living with COPD). The PPI group confirmed their suggestions had been addressed. The steering group confirmed the finalised PtDA was comprehensible, easy to use and acceptable to the target population. It scored 9/11 for adherence to the user-centred design process (see Supplementary Materials, S5 in S1 File; [41]).

**Step 6: Supporting intervention components.**   Results from the alpha testing, design steps 2b and 3 (see Supplementary materials, S2 in S1 File), and PtDA and decision coaching literature [28, 63] indicated that to successfully implement SDM into practice there was a need to train PR HCPs how to embed SDM and use a PtDA during their interaction with individuals. It also highlighted the need for standardising the SDM consultation so that it would be delivered with equity. Therefore, decision coaching training and a consultation prompt were chosen as supporting intervention components.

## Decision coaching training and consultation prompt

Synthesis of the guidance collated for the development of the teaching materials and steering group recommendations led to the inclusion of practical teaching methods with the opportunity for feedback (i.e. role play), activities to distinguish between fast (intuitive) and slow (rational) thinking and between patient-centred care and SDM, relevant scenarios for PR HCPs to reflect upon using the three-talk model of SDM [29], an introduction on how to use a PtDA, a ready-made consultation prompt, and the use of a post-teaching knowledge test to consolidate the learning.

Following review by the steering group, the teaching materials were expanded to include an activity for HCPs to personalise their consultation prompt and videos to demonstrate SDM within the PR context.

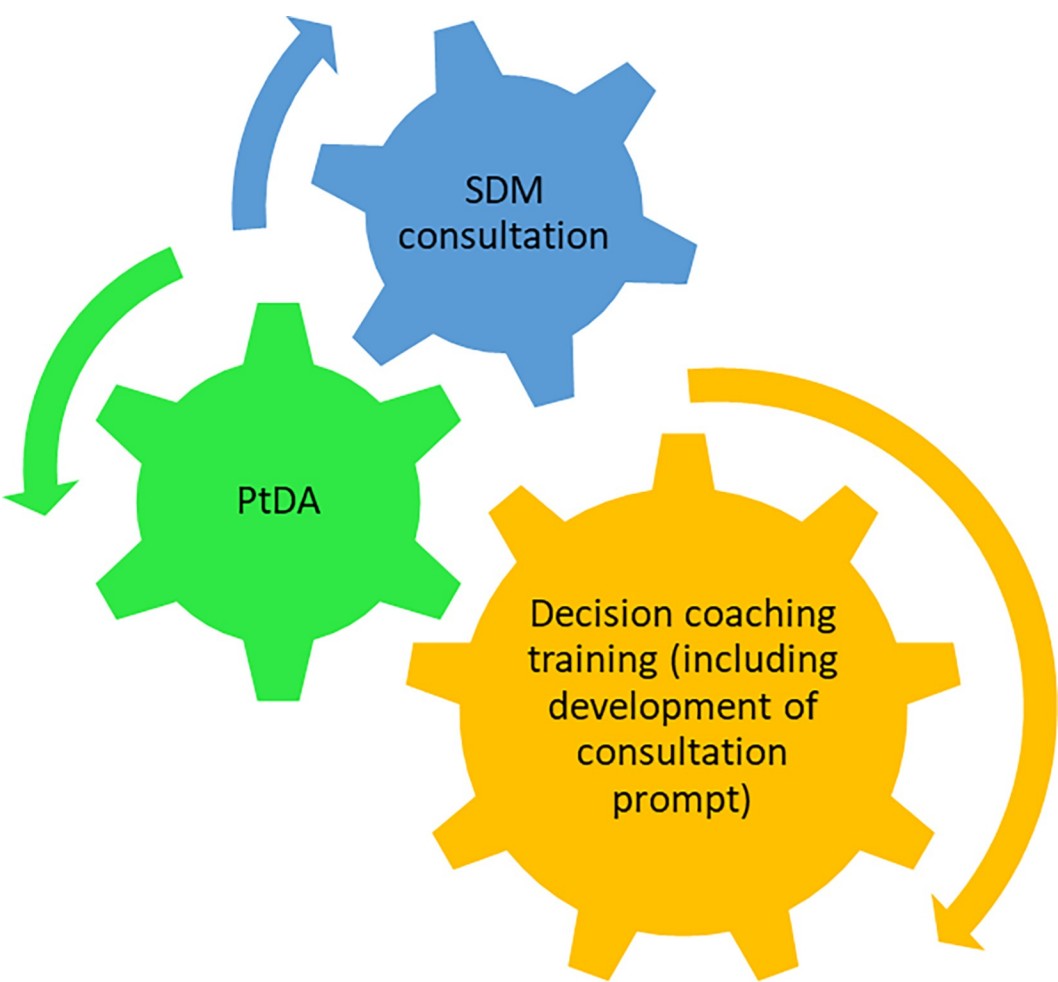

**Fig 4. Finalised SDM intervention components.** SDM = shared decision making, PtDA = patient decision aid.

The finalised training consisted of a bespoke 2-hour workshop specifically relevant to SDM in PR with an informal feedback and reflection session 1 month later. The decision coaching session plan is provided elsewhere [44].

## The SDM intervention

The finalised SDM intervention consisted of three distinct components: decision coaching training for PR HCPs, the PtDA, and a SDM consultation which was guided by a consultation prompt. This was individually developed by the PR HCPs during the decision coaching training (Fig 4) and consisted of individualised prompts and conversation starters to guide the SDM consultation. All components are protected by copyright law. A full description of intervention delivery is detailed in the protocol [44].

## Discussion

We have described the rigorous methodological development of a three component SDM intervention to support individuals living with COPD and their PR HCP make a decision about the menu of PR programmes. This is the first intervention to address the unmet support needs of individuals and HCPs when embedding SDM into the PR pathway and it is timely

following the increased need for a menu of PR programmes following the Coronavirus Disease 2019 pandemic.

We utilised three overarching theoretical frameworks; the MRC complex intervention development and evaluation framework [40], the MIND-IT model [31] and the ODSF [45] alongside additional tools and frameworks which provided specific guidance on PtDA and supporting intervention component development. Development considered implementation throughout to support ease of integration into practice should it prove both effective and acceptable.

Alpha testing of the PtDA with a PPI group and PR HCPs was conducted to identify any acceptability, usability, and comprehensibility support needs and ensure that the intervention could structurally fit into routine care. This was to increase the likelihood of the intervention being delivered with fidelity. All agreed on the value of the PtDA and the need to develop supporting components to ensure SDM implementation, integration, and thereby observability when measuring intervention fidelity. This led to the development of a bespoke decision coaching training session which embedded an activity to support PR HCPs create a personalised consultation prompt to guide their SDM consultations. This combination of components has previously been proposed to support the adoption of SDM in practice [27] and increase individuals' knowledge [64]. Whilst there is very limited research into the barriers of SDM within PR decision-making, insufficient knowledge of SDM amongst HCPs has been cited and so our intervention aligns with Jiang and colleagues' recommendations to provide PR HCPs with training in SDM [65]. That said gaining more knowledge on SDM may not necessarily lead to behaviour change, therefore the link between increased knowledge and taking action should be addressed/ measured in future research.

Regarding acceptability, we found that HCPs were concerned about the time it would take to conduct SDM with individuals using the PtDA. Whilst this is not a new concern [66], this feedback led us to decide that clearer instruction would be needed to inform individuals on our expectations of their engagement with the PtDA prior to their SDM consultation with a PR HCP. Interestingly, the time burden was not an area that patients identified as a barrier. It may be that the patients would relish the additional time to discuss PR option with their healthcare provider, whereas clinicians were concerned about including the SDM intervention within an already time-pressured assessment period.

For the comprehensibility, individuals and HCPs largely agreed that more information was required to illustrate the attributes of each option. Unmet information needs have been commonly cited as a barrier to PR decision-making [20] and were a key reason for us to embed SDM within the PR pathway [67], hence these became the largest modifications made to the PtDA. Varying viewpoints were observed regarding certain terminology within the PtDA and the use of scientific data and references. Since we met with PR HCPs first, we were able to take their concerns to the PPI group to confirm which terms to change and which to retain. This enabled us to maintain a readability score appropriate for individuals living with COPD [49–51] but also ensure the terminology used was familiar and within their vocabulary. There were proposals regarding its comprehensibility that the steering group decided not to action. For example, PR HCPs suggested the addition of illustrative pictures to each option. This was discussed during the early stages of prototype development and reappraisal of this proposal yielded the same response from the steering group. They felt that each option could not be fully illustrated in a single picture. We were informed by Martin and colleagues' guidance on the presentation of information, who stated that data needs to be balanced and without bias towards a single option [48]. All of our decisions on comprehensibility were to ensure that the PtDA provided all information necessary to increase individuals health literacy and thereby facilitate their informed decision-making for PR [47, 51].

For the usability of the PtDA, all individual and PR HCP proposals were actioned. This led the PtDA to have clearer instructions, additional prompts to encourage support from others in their decision-making, activities that were easier to engage with, and a more digestible format. This change facilitated the presentation of options side by side, especially once the additional attributional data had been added, which importantly helped to remove bias towards one option over another [48].

Key strengths of intervention development were that it was developed by a multidisciplinary expert team, including representation from individuals living with COPD, and it took a clear systematic and transparent approach. A limitation was two authors from the telerehabilitation systematic review could not be contacted which meant their data could not be included within the PtDA. This omission may have resulted in an overall over or underestimation of the efficacy of the home-based PR options. In our alpha testing phase, we believe we had good representation from PR HCPs as the majority of our local team contributed, however, we recognise that we had a small number of individuals who contributed. It may be that we did not capture the diversity of opinions of all individuals living with COPD, however we did obtain rich, in depth data from this qualitative work We will be exploring individual appraisals further in a mixed methods feasibility study. Finally, the intervention as it stands is only available in English. The steering group will consider its cross-cultural adaptation following its feasibility and acceptability evaluation.

## Conclusions

We have developed a three-component SDM intervention to support individuals living with COPD make informed and values-based decision about PR using a systematic, research, user-centred and co-design process. We considered implementation and integrational ability throughout by conducting pre-feasibility testing with individuals living with COPD and PR HCPs. Should it prove acceptable and effective clear implementation strategies have been outlined to support its integration into the PR pathway.

## Study registration

Registered on Clinical Trials.gov (NCT04990180) in August 2021. Protocol version 7.0, 9 November 2021.

## Supporting information

**S1 File.**
(DOCX)

## Acknowledgments

We would like to thank all participants for their valued contribution to this research.

## Author Contributions

**Conceptualization:** Amy C. Barradell.

**Data curation:** Amy C. Barradell.

**Formal analysis:** Amy C. Barradell, Hilary L. Bekker, Linzy Houchen-Wolloff, Noelle Robertson, Sally J. Singh.

**Funding acquisition:** Sally J. Singh.

**Methodology:** Amy C. Barradell, Hilary L. Bekker.

**Project administration:** Amy C. Barradell.

**Supervision:** Hilary L. Bekker, Linzy Houchen-Wolloff, Kim Marshall-Nichols, Noelle Robertson, Sally J. Singh.

**Validation:** Hilary L. Bekker, Linzy Houchen-Wolloff, Kim Marshall-Nichols, Noelle Robertson, Sally J. Singh.

**Writing – original draft:** Amy C. Barradell.

**Writing – review & editing:** Amy C. Barradell, Hilary L. Bekker, Linzy Houchen-Wolloff, Kim Marshall-Nichols, Noelle Robertson, Sally J. Singh.

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
