## [Decision Letter · Decision Letter 0]

5 Mar 2024

PONE-D-23-33990A Shared Decision-Making intervention for individuals living with COPD who are making Pulmonary Rehabilitation treatment options with respiratory and physiotherapy healthcare professionals: An iterative development process with qualitative methodsPLOS ONE

Dear Dr. Houchen-Wolloff,

Thank you for submitting your manuscript to PLOS ONE. After careful consideration, we feel that it has merit but does not fully meet PLOS ONE’s publication criteria as it currently stands. Therefore, we invite you to submit a revised version of the manuscript that addresses the points raised during the review process.

We look forward to receiving your revised manuscript.

Kind regards,

Tamer I. Abo Elyazed, Ph.d

Guest Editor

PLOS ONE

Journal Requirements:

3. PLOS requires an ORCID iD for the corresponding author in Editorial Manager on papers submitted after December 6th, 2016. Please ensure that you have an ORCID iD and that it is validated in Editorial Manager. To do this, go to ‘Update my Information’ (in the upper left-hand corner of the main menu), and click on the Fetch/Validate link next to the ORCID field. This will take you to the ORCID site and allow you to create a new iD or authenticate a pre-existing iD in Editorial Manager. Please see the following video for instructions on linking an ORCID iD to your Editorial Manager account: https://www.youtube.com/watch?v=_xcclfuvtxQ.

Additional Editor Comments (if provided):

The title of the manuscript needs to be re structured to clarify the goal of the study in more simple abbreviated words; e.g I suggest [A shred decision making between health care professionals and individuals undergoing Pulmonary rehabilitation : An iterative development process with qualitative methods].

Reviewers' comments:

Reviewer's Responses to Questions

**Comments to the Author**

1. Is the manuscript technically sound, and do the data support the conclusions?

Reviewer #1: Yes

Reviewer #2: Yes

2. Has the statistical analysis been performed appropriately and rigorously? 

Reviewer #1: N/A

Reviewer #2: N/A

3. Have the authors made all data underlying the findings in their manuscript fully available?

Reviewer #1: Yes

Reviewer #2: Yes

4. Is the manuscript presented in an intelligible fashion and written in standard English?

Reviewer #1: Yes

Reviewer #2: Yes

5. Review Comments to the Author

Reviewer #1: This article outlines the development of an intervention model for shared decision making (SDM) for individuals with Chronic Obstructive Pulmonary Disease (COPD) making decisions about Pulmonary Rehabilitation (PR). The intervention, consisting of a Patient Decision Aid (PtDA), training of healthcare professionals (HCPs) in decision coaching, and a consultation prompt, was developed over a nine-month period. The study utilized iterative development processes, including forming a steering group, defining the scope of the PtDA, data collection, prototype development, and alpha-testing with COPD individuals and PR HCPs. The PtDA was revised six times to ensure acceptability. The supporting components, decision coaching, and a consultation prompt were added to empower PR HCPs in SDM. The final intervention aims to facilitate informed and values-based decisions about PR for individuals with COPD.

This article is well-written, and the authors behind it possess solid expertise in the field. Therefore, the article has numerous strengths that enhance its academic rigor and contribute to the professional interest in this work.

These include:

1.Comprehensive Frameworks: The intervention development process utilized broad overarching frameworks, including the MRC complex intervention development and evaluation framework, the MIND-IT model, and the Ottawa Decision Support Framework. This ensured a comprehensive and methodologically sound approach.

2.Iterative Development/proto-typing: The PtDA underwent six revisions based on alpha-testing with both individuals with COPD and PR HCPs. This iterative process strengthened the intervention's clarity, comprehensibility, and usability.

3.Clear Scope: The intervention's scope was well-defined, aiming to standardize discussions between individuals and PR HCPs, facilitating information exchange, deliberation, and decision-making.

4.User-Centered Design: The PtDA development adhered to a user-centered design checklist, ensuring that the final product was tailored to the needs and preferences of the target population.

5.Integration of Supporting Components: Recognizing the need for additional support, decision coaching training for PR HCPs and a consultation prompt were integrated, addressing concerns about the time required for SDM consultations. This is extremely important, as most Shared Decision-Making (SDM) experts both know and acknowledge that Patient Decision Aids (PtDAs) cannot stand alone and are not a tick-a-box solution for SDM implementation.

6.Readability Considerations: The PtDA's readability was assessed, and modifications were made to ensure appropriateness for individuals living with COPD, contributing to improved health literacy.

However, there are also several limitations, many of which could be addressed to enhance the content, scientific rigor, and reader interest of this article. These include (read as reviewer's comments):

1.Limited Uptake Data: The intervention was completed two years ago, yet there is a lack of data on the uptake of the three interventions. While it is acknowledged that the authors might intend to publish this in a separate article, the absence of uptake data in the current study limits a comprehensive assessment of the intervention's real-world impact.

2.Small Alpha Test Sample: The alpha-testing phase involved a small number of participants, with 7 (8) healthcare professionals and 3 (4) patients. The limited size of the alpha test group raises questions about the generalizability and robustness of the intervention, particularly in capturing a diverse range of perspectives.

3.Absence of Beta Testing Data: The study lacks beta testing data conducted in real-life conditions (field tests) about feasibility.

In the PtDA development model utilized by the authors, based on Coulter et al.'s 2013 article, Coulter et al. have outlined both alpha testing, focusing on acceptability and usability, and beta testing, which includes real-world field testing for feasibility. However, the presentation lacks data or reflections on why only alpha test findings are presented, and whether the PtDA development process concluded after alpha testing without progressing to beta testing. It is possible that the authors chose to deviate from the model, adopted a different approach, or pursued an alternative path, but there is no available data or reflections on this matter. Therefore it appears, that the PtDAs appear to lack comprehensive field testing with patients or evaluation by clinicians not involved in the development process, undermining the understanding of its feasibility in practical settings.

4.Timeline Discrepancy: Reference is made to the use of the user-centered design checklist (reference 41) at the early stages of the study, during the intial phase of outlining the scope of the intervention. However, the timeline raises concerns, as reference 41 was published in October 2021, several months after the project's initiation in March 2021 (although reference 41 was online mid-June this is still several months after the start of this project)? So was this incorporated semi-retrospective?

5.Sequential Figure Reference: On page 7 of the manuscript, Figure 2 is referenced before Figure 1. Typically, figures and tables should be referenced consecutively in a scientific article. Maybe this edit will be addressed during the proofreading process to ensure consistency and compliance with common scientific standards.

6.Imbalance in Steering Group Composition: The steering group comprises 5 specialists/academics/experts and only 1 patient. This composition may not sufficiently address the inherent power imbalance, potentially limiting the representation of the patient's voice in decision-making. This should be reflected on and argued why only one patient representative was included.

7.Unclear Scope of Consultation Prompt: The manuscript lacks information about the scope, appearance, and specific content of the consultation prompt. In my view, this part of the intervention is too implicit, and it is unclear to readers what specifically it entails. A more detailed explanation is needed to understand the intervention's purpose and what exactly this is in the context of the study.

8.Limited Display of PtDA: Only the front page of the PtDA is presented in S4: “Finalised PtDA (preview; version 6). “To provide a comprehensive understanding, the entire PtDA should be showcased, as the front page alone does not convey sufficient information about its contents.

Reviewer #2: The paper describes the process of developing a shared decision making (SDM) intervention to enhance PR uptake in patients living with COPD.

It is clearly an important area, and the authors explain the purpose of the intervention well. The main suggestion I have would be to provide more information regarding the findings and details of the intervention itself within the main body of the paper instead of as supplemental materials. Readers have a tendency to focus more on the paper as opposed to accessing the supplemental materials.

I have summarised specific suggestions for further improvement to the manuscript below:

Abstract

•May have missed this, but a ‘useable leaflet’ only appears to be referenced in the abstract, with no mention of what this looks like elsewhere within the paper. It would be good to expand.

Introduction

•The authors provide a comparison of an online PR programme vs. traditional programme, but more information about what a traditional programme looks like would be beneficial. Readers of PLOS ONE typically come from different backgrounds and may not be familiar. Would be good to provide more context to the area.

•Similarly, commonly cited barriers to the uptake of traditional PR (e.g. organisational constraints) are mentioned, but an expansion and explanation of these barriers would be good.

Methods

•Step 4 describes an exercise used to support individuals identifying core values and beliefs, but a description of what this entailed would be beneficial.

•A ‘think-aloud approach’ is described in the Alpha testing section. I wonder if the authors have any quotes they could present to support their findings.

Results

•It is interesting that the HCPs seemed to identify burden as one area that required attention, but the PPI group did not. I wonder if the authors had any thoughts as to why this was the case? Would be interesting to discuss.

Discussion

•This section summarises the process of developing the SDM intervention quite nicely.

•In one area, the authors talk about limited research on the barriers of SDM, with one being insufficient knowledge of SDM amongst HCPs. The authors could explore whether gaining more knowledge always leads to taking action. In the field of behaviour change, it has been observed that knowing what to do does not always result in actual behaviour change. It would be valuable for the authors to contemplate this aspect in the context of potential future research.

6. PLOS authors have the option to publish the peer review history of their article (what does this mean?). If published, this will include your full peer review and any attached files.

Reviewer #1: No

Reviewer #2: **Yes: **Louisa Lawrie

---

## [Author Response · Author response to Decision Letter 0]

3 May 2024

Please see attached file for detailed response

---

## [Decision Letter · Decision Letter 1]

5 Jul 2024

PONE-D-23-33990R1A shared decision-making intervention between health care professionals and individuals undergoing Pulmonary rehabilitation: An iterative development process with qualitative methodsPLOS ONE

Dear Dr. Houchen-Wolloff,

Thank you for submitting your manuscript to PLOS ONE. After careful consideration, we feel that it has merit but does not fully meet PLOS ONE’s publication criteria as it currently stands. Therefore, we invite you to submit a revised version of the manuscript that addresses the points raised during the review process. Please submit your revised manuscript by Aug 19 2024 11:59PM. If you will need more time than this to complete your revisions, please reply to this message or contact the journal office at plosone@plos.org. Please include the following items when submitting your revised manuscript:A rebuttal letter that responds to each point raised by the academic editor and reviewer(s). You should upload this letter as a separate file labeled 'Response to Reviewers'.A marked-up copy of your manuscript that highlights changes made to the original version. You should upload this as a separate file labeled 'Revised Manuscript with Track Changes'.An unmarked version of your revised paper without tracked changes. You should upload this as a separate file labeled 'Manuscript'.If applicable, we recommend that you deposit your laboratory protocols in protocols.io to enhance the reproducibility of your results. Protocols.io assigns your protocol its own identifier (DOI) so that it can be cited independently in the future. For instructions see: https://journals.plos.org/plosone/s/submission-guidelines#loc-laboratory-protocols. Additionally, PLOS ONE offers an option for publishing peer-reviewed Lab Protocol articles, which describe protocols hosted on protocols.io. Read more information on sharing protocols at https://plos.org/protocols?utm_medium=editorial-email&utm_source=authorletters&utm_campaign=protocols.

We look forward to receiving your revised manuscript.

Kind regards,

Tamer I. Abo Elyazed, Ph.d

Guest Editor

PLOS ONE

Journal Requirements:

Reviewers' comments:

Reviewer's Responses to Questions

**Comments to the Author**

1. If the authors have adequately addressed your comments raised in a previous round of review and you feel that this manuscript is now acceptable for publication, you may indicate that here to bypass the “Comments to the Author” section, enter your conflict of interest statement in the “Confidential to Editor” section, and submit your "Accept" recommendation.

Reviewer #1: (No Response)

Reviewer #2: All comments have been addressed

2. Is the manuscript technically sound, and do the data support the conclusions?

Reviewer #1: Yes

Reviewer #2: Yes

3. Has the statistical analysis been performed appropriately and rigorously? 

Reviewer #1: N/A

Reviewer #2: N/A

4. Have the authors made all data underlying the findings in their manuscript fully available?

Reviewer #1: Yes

Reviewer #2: Yes

5. Is the manuscript presented in an intelligible fashion and written in standard English?

Reviewer #1: Yes

Reviewer #2: Yes

6. Review Comments to the Author

Reviewer #1: Thank you for the sufficient response to the raised criticism. I believe the article is ready for publication. However, I don't think it is clear to the reader when reading the figure legends for supplementary figure 4 that it is only a small excerpt of figure 4 that is presented in the supplementary material. I find it unfortunate that they do not display the entire PtDA in supplementary figure 4. However, if the authors are concerned due to copyright issues, it must be taken into account. Nevertheless, they should explain to the reader that only a small excerpt is shown in the supplementary material, for example, that it is page 1, x, and x out of a total of xx pages of the paper-based (?) PtDA.

Reviewer #2: (No Response)

7. PLOS authors have the option to publish the peer review history of their article (what does this mean?). If published, this will include your full peer review and any attached files.

Reviewer #1: No

Reviewer #2: **Yes: **Louisa Lawrie

---

## [Author Response · Author response to Decision Letter 1]

8 Jul 2024

08.07.24

Dear Tamer I. Abo Elyazed, Ph.d

RE: PONE-D-23-33990 Revision 2

A Shared Decision-Making intervention for individuals living with COPD who are making Pulmonary Rehabilitation treatment options with respiratory and physiotherapy healthcare professionals: An iterative development process with qualitative methods

PLOS ONE

Thank you for the opportunity to revise our manuscript. We have responded to the remaining reviewer comment below and have uploaded a tracked changes and final copy of the revised manuscript.

Reviewer #1: Thank you for the sufficient response to the raised criticism. I believe the article is ready for publication. However, I don't think it is clear to the reader when reading the figure legends for supplementary figure 4 that it is only a small excerpt of figure 4 that is presented in the supplementary material. I find it unfortunate that they do not display the entire PtDA in supplementary figure 4. However, if the authors are concerned due to copyright issues, it must be taken into account. Nevertheless, they should explain to the reader that only a small excerpt is shown in the supplementary material, for example, that it is page 1, x, and x out of a total of xx pages of the paper-based (?) PtDA.

In the manuscript- it is made clear that only a small excerpt is shown. 

In the supplement- the following text has been added: ‘S4: Finalised PtDA (preview; pages 1, 7 and 13 of 17 page booklet. Subject to copyright version 6)’

We look forward to hearing from you in due course.

Yours sincerely,

Linzy Houchen-Wolloff PhD

Corresponding author on behalf of all co-authors.

---

## [Editor Report · Decision Letter 2]

10 Jul 2024

A shared decision-making intervention between health care professionals and individuals undergoing Pulmonary rehabilitation: An iterative development process with qualitative methods

PONE-D-23-33990R2

Dear Dr. Houchen-Wolloff,

We’re pleased to inform you that your manuscript has been judged scientifically suitable for publication and will be formally accepted for publication once it meets all outstanding technical requirements.

Kind regards,

Tamer I. Abo Elyazed, Ph.d

Guest Editor

PLOS ONE
---

## [Editor Report · Acceptance letter]

7 Aug 2024

PONE-D-23-33990R2 

PLOS ONE

Dear Dr. Houchen-Wolloff, 

I'm pleased to inform you that your manuscript has been deemed suitable for publication in PLOS ONE. Congratulations! Your manuscript is now being handed over to our production team.

Kind regards, 

on behalf of

Dr. Tamer I. Abo Elyazed 

Guest Editor

PLOS ONE